# Biomarkers and Seaweed-Based Nutritional Interventions in Metabolic Syndrome: A Comprehensive Review

**DOI:** 10.3390/md22120550

**Published:** 2024-12-04

**Authors:** Ana Valado, Margarida Cunha, Leonel Pereira

**Affiliations:** 1Polytechnic University of Coimbra, Coimbra Health School, Biomedical Laboratory Sciences, Rua 5 de Outubro—S. Martinho do Bispo, Apartado 7006, 3045-043 Coimbra, Portugal; margaridacunha2002@hotmail.com; 2Research Centre for Natural Resources Environment and Society (CERNAS), Polytechnic University of Coimbra, Bencanta, 3045-601 Coimbra, Portugal; 3MARE—Marine and Environmental Sciences Centre/ARNET-Aquatic Research Network, University of Coimbra, 3000-456 Coimbra, Portugal; 4H&TRC—Health & Technology Research Center, Coimbra Health School, Polytechnic University of Coimbra, Rua 5 de Outubro, 3045-043 Coimbra, Portugal; 5Centre for Functional Ecology: Science for People & Planet, Marine Resources, Conservation and Technology—Marine Algae Lab, Department of Life Sciences, University of Coimbra, 3000-456 Coimbra, Portugal; leonel.pereira@uc.pt

**Keywords:** metabolic syndrome, obesity, cardiovascular disease, biomarkers, therapy, marine algae

## Abstract

Metabolic Syndrome (MetS) is a complex, multifactorial condition characterized by risk factors such as abdominal obesity, insulin resistance, dyslipidemia and hypertension, which significantly contribute to the development of cardiovascular disease (CVD), the leading cause of death worldwide. Early identification and effective monitoring of MetS is crucial for preventing serious cardiovascular complications. This article provides a comprehensive overview of various biomarkers associated with MetS, including lipid profile markers (triglyceride/high-density lipoprotein cholesterol (TG/HDL-C) ratio and apolipoprotein B/apolipoprotein A1 (ApoB/ApoA1) ratio), inflammatory markers (interleukin 6 (IL-6), tumor necrosis factor alpha (TNF-α), plasminogen activator inhibitor type 1 (PAI-1), C-reactive protein (CRP), leptin/adiponectin ratio, omentin and fetuin-A/adiponectin ratio), oxidative stress markers (lipid peroxides, protein and nucleic acid oxidation, gamma-glutamyl transferase (GGT), uric acid) and microRNAs (miRNAs) such as miR-15a-5p, miR5-17-5p and miR-24-3p. Additionally, this review highlights the importance of biomarkers in MetS and the need for advancements in their identification and use for improving prevention and treatment. Seaweed therapy is also discussed as a significant intervention for MetS due to its rich content of fiber, antioxidants, minerals and bioactive compounds, which help improve cardiovascular health, reduce inflammation, increase insulin sensitivity and promote weight loss, making it a promising nutritional strategy for managing metabolic and cardiovascular health.

## 1. Introduction

Metabolic Syndrome (MetS) was first described in 1988 as “Syndrome X”. This condition is characterized by the combination of at least three metabolic abnormalities, which include increased abdominal circumference, fasting blood glucose, blood pressure and triglycerides (TG); decreased high-density lipoprotein cholesterol (HDL-C); and the development of pathologies such as type 2 diabetes mellitus (T2DM) and cardiovascular diseases (CVD). Compared to individuals without MetS, this condition doubles the risk of developing CVD in the following 5 to 10 years and is also associated with a fivefold increase in the risk of developing T2DM [1,2]. According to the atlas of the International Diabetes Federation [3], the global prevalence of diabetes is expected to rise to 10.4% by 2040. There is no similar global data on MetS, as it is more complicated to measure, but as MetS is around three times more common than diabetes, the global prevalence can be estimated at around a quarter of the world’s population [2]. With the growing incidence of MetS, it is crucial to recognize the importance of early diagnosis, underlining the need to study new possible biomarkers allowing for more effective preventive interventions.

In recent years, seaweed has emerged as a promising nutritional intervention in the management of MetS. Due to its rich composition in fiber, polyphenols, polyunsaturated fatty acids and various bioactive compounds, seaweed has shown potential to positively modulate biomarkers associated with MetS, such as glycemia, lipid profile and inflammation. In addition, its antioxidant and anti-inflammatory action offers significant benefits for the prevention of metabolic complications. As part of a functional food approach, algae can complement current strategies for the prevention and treatment of MetS, providing a natural, affordable and safe alternative to traditional pharmacological interventions [4].

This article comprehensively reviews the relevant biomarkers of MetS and explores how algae-based nutritional interventions can positively impact these parameters, offering new perspectives for the management of this complex condition.

## 2. Pathogenesis of Metabolic Syndrome

MetS is a term given to a set of conditions, including hyperglycemia, atherogenic dyslipidemia, insulin resistance (IR), systemic arterial hypertension and central/abdominal obesity. Recognized by the World Health Organization (WHO) as a Chronic Non-Communicable Disease (CNCD) [5], MetS has been associated with CVD which, in turn, is correlated with the progression of atherosclerosis in response to chronic inflammation and vascular endothelial dysfunction and T2DM, mainly due to disturbances in insulin secretion, action and IR [6].

To diagnose MetS, an individual is considered to have the condition when there are changes in at least three of the following parameters: waist circumference (≥102 cm for men and ≥88 cm for European women), high TG levels (≥150 mg/dL), reduced HDL-C (<40 mg/dL for men and <50 mg/dL for European women), high fasting glucose (≥100 mg/dL) and high blood pressure (≥130/85 mmHg) [5,7,8].

Although the pathophysiology of MetS has not yet been fully elucidated, several factors have been described as predisposing a person to its development, such as genetic and epigenetic predisposition, smoking, obesity, increasing age, sedentary lifestyle, inadequate diet, excessive alcohol consumption, low socioeconomic status, as well as other lifestyle aspects [7,9,10].

The most plausible hypothesis for the pathophysiology of MetS is IR. Insulin is a hormone secreted by pancreatic beta cells in response to high blood glucose levels. Insulin will inhibit lipolysis and hepatic gluconeogenesis while increasing glucose uptake in the liver, muscles and adipose tissue. When IR develops in adipose tissue, insulin-mediated inhibition of lipolysis is impaired. This results in an increase in circulating free fatty acids (FFAs), exacerbating IR and triggering changes in the insulin signaling cascade in different organs, thus establishing a vicious cycle. In muscles, FFAs interfere with the activity of phosphatidylinositol 3-kinase (P13K) associated with the insulin receptor substrate 1 (IRS-1), decreasing the translocation of GLUT-4 to the cell surface and, consequently, reducing glucose uptake. At the same time, FFAs act on the liver, promoting both gluconeogenesis and lipogenesis. As a result, a hyperinsulinemic state occurs to maintain normal glucose levels. However, this compensation eventually fails, resulting in a decrease in insulin levels, aggravated by the lipotoxic effect of FFAs on pancreatic beta cells [6].

Lifestyles with a high calorie intake and a lack of regular exercise are considered responsible for triggering MetS. In this sense, visceral adiposity is one of the first triggers of MetS and can lead to obesity and disorders in adipose tissue [11]. The presence of disturbances at the cellular level causes a decrease in the tissue’s response to insulin stimulation, which is characterized by failures in glucose uptake and oxidation. This imbalance causes chronic hyperglycemia, which in turn triggers oxidative stress (OS) and an inflammatory response that causes cell damage. IR leads to the development of dyslipidemia with high TG levels, low HDL-C levels and an increase in low-density lipoprotein cholesterol (LDL-C). This lipid triad accompanied by endothelial dysfunction can result in atherosclerotic events [6].

Adipose tissue not only is an energy storage site but also acts as an active endocrine organ that secretes a variety of bioactive molecules, known as adipokines, which include leptin, adiponectin, inflammatory cytokines, resistin and visfatin [12,13]. Dysregulation of certain adipokines can promote pathogenic conditions associated with obesity, lipid accumulation and IR [14].

An emerging metabolic risk factor is a pro-inflammatory state, and an excess of adipose tissue can contribute to this same factor given that it leads to the accumulation of macrophages. Macrophages are key components of the immune system that reside in adipose tissue. During obesity, there is an influx of pro-inflammatory macrophages into the adipose tissue, leading to the formation of crowns of fat bodies and the establishment of a chronic inflammatory state. These macrophages together with other immune cells, such as T cells, play a central role in promoting IR and metabolic dysfunction associated with MetS [15]. The evolution of MetS triggered by the various pathogenic pathways culminates in a pro-inflammatory state that clarifies the increase in inflammatory markers such as interleukin 6 (IL-6), C-reactive protein (CRP) and tumor necrosis factor alpha (TNF-α) [16].

Chronic inflammation induced by the interaction between adipocytes and macrophages contributes to IR, a central component of MetS. The activation of inflammatory signaling pathways, such as the nuclear factor kappa B (NF-κB) pathway, inhibits insulin signaling in target tissues such as the liver, skeletal muscle and adipose tissue, resulting in an increase in glycemia and dyslipidemia [16].

The pathophysiology of MetS is extremely complex and is still not fully understood. Studies suggest that an imbalance between oxidative and antioxidant processes may play a significant role in the development of this condition. Increased OS biomarkers and decreased antioxidant defenses have been observed in the blood plasma of MetS patients, suggesting an increased in vivo production of Reactive Oxygen Species (ROS). Specifically, a decrease in antioxidant capacity has been reported, manifested by reduced concentrations of vitamin C and α-tocopherol in serum, as well as a decrease in the activity of the Superoxide Dismutase (SOD) enzyme and an increase in levels of protein and lipid oxidation [17].

OS arises due to the disproportionate production of ROS and Reactive Nitrogen Species (RNS), in association with a decrease in the quantity, expression and compromised activity of antioxidant systems. In appropriate concentrations, ROS and RNS perform functions as signaling molecules, driving cellular activities and providing cellular protection. However, in conditions of excess, as observed in inflamed tissues, these species can generate other highly reactive molecules capable of irreversibly oxidizing proteins, lipids and nucleic acids. It is crucial to note the oxidative modification of critical enzymes or regulatory sites, whose redox alteration triggers changes in cell signaling and the activation of programmed cell death [18].

The prevalent inflammatory profile described in MetS is not the result of tissue damage, infection or autoimmune response but of meta-inflammation. Meta-inflammation is a chronic state of inflammation characterized by the dysregulation of pro-inflammatory cytokines and chemokines and mediated by macrophages. Recent studies have highlighted the importance of adipose tissue-resident macrophages as the main source for meta-inflammation, and miRNAs are key molecules in these signaling pathways [18].

Adipose tissue is one of the main sources of circulating miRNAs, recently described as a new form of adipokines. Several adipose tissue-derived miRNAs are deeply associated with adipocyte differentiation and have been identified with an essential role in obesity-associated inflammation, IR and tumor microenvironment. During obesity, adipose tissue can completely alter the profile of secreted miRNAs, influencing circulating miRNAs and impacting the development of different pathological conditions, such as obesity, MetS and cancer [7].

MiRNAs are small non-coding RNA molecules that play a suppressive role in gene expression by binding to the three-prime untranslated region (3′UTR) of the target messenger ribonucleic acid (mRNA). These molecules can be present in various tissues and body fluids, such as saliva, plasma, serum and whole blood, being encapsulated in extracellular vesicles (EVs) or as circulating miRNA. A disturbance in miRNA expression affects gene expression, thus affecting cell function in the pathological process. Identifying miRNAs with abnormal expression in tissue or serum may be important for the early diagnosis and treatment of this condition [19].

Changes in the profiles of miRNAs present in circulating fluids and tissues have a direct impact on the physiology of cells and tissues involved in glucose and lipid metabolism, such as pancreatic β-cells, hepatocytes, skeletal muscle and adipose tissue. These changes modulate mRNA translation. Due to the ease with which miRNA-rich samples such as blood, plasma and saliva can be obtained, assessing the associations between miRNA levels and diseases can lead to the development of diagnostics and prognostic markers with high sensitivity and specificity [20].

The genetic susceptibility of MetS has increasingly attracted the attention of researchers with the aim of providing a better understanding of MetS. Due to their role in regulating gene expression and their association with metabolic processes relevant to MetS, miRNAs have the potential to serve as useful biomarkers for the detection, prognosis and monitoring of MetS. The identification of specific miRNA expression patterns associated with MetS can provide important insights into the pathophysiology of MetS and help with risk stratification, personalization of treatments and the development of new therapeutic approaches [18]. In this sense, miRNAs represent a promising area of research as biomarkers in MetS.

Conventional drug therapies for MetS focus mainly on treating dysglycemia, dyslipidemia and hypertension separately. However, several side effects of these drugs have been reported, including depressed mood and anxiety. Consequently, there has been a growing interest in natural treatments that offer benefits with minimal side effects.

Instead of interventions based exclusively on calorie restrictions or individual nutrients, the use of functional food-based dietary interventions is recommended for the prevention and treatment of MetS [21]. In this context, seaweed, also known as macroalgae, and its bioactive components have attracted significant attention as potential functional foods for the management of MetS in humans.

## 3. Metabolic Syndrome and Some Associated Pathologies

### 3.1. Cardiovascular Diseases

CVD is the leading cause of mortality and morbidity worldwide, and it is estimated that annual mortality will rise to almost 24 million by 2030 [22]. They are a group of pathologies that affect the heart and blood vessels, including conditions such as coronary artery disease, cerebrovascular disease, congestive heart failure and peripheral vascular disease. They are chronic diseases that progress gradually throughout life and remain asymptomatic for a long time. Individuals with MetS have a higher risk of developing CVD compared to those without MetS over the next 5–10 years, and the long-term risk is even higher [1,2]. IR, a key component of MetS, is associated with metabolic abnormalities that contribute to the development and progression of CVD. IR promotes endothelial dysfunction, chronic inflammation, atherogenic dyslipidemia and hypertension, all important risk factors for CVD [6].

In addition, abdominal obesity, present in MetS, is strongly associated with an adverse lipid profile, including high TG levels and reduced LDL-C, which increases the risk of atherosclerosis and cardiovascular events [23].

### 3.2. Diabetes Mellitus Type 2

MetS has been associated with T2DM due to its high prevalence worldwide, being related to the increase in obesity and a sedentary lifestyle. It is estimated that 578 million people will have diabetes by 2030 and 700 million by 2045 [24]. Studies suggest that individuals with MetS are five times more likely to develop T2DM [25]. The onset of T2DM is one of the main comorbid conditions associated with MetS. In MetS, IR plays a central role, resulting from the direct inhibition of insulin receptors in cells by inflammatory cytokines released by adipose tissue. This phenomenon compromises the intracellular transport of glucose, which, over time, contributes to the development of T2DM in individuals with MetS [15,24].

Individuals with T2DM are at greater risk of micro- and macrovascular complications due to hyperglycemia and individual components of IR. Thus, the pathophysiology of T2DM involves a variable combination of IR and a decrease in insulin secretion by pancreatic beta cells [24].

### 3.3. Non-Alcoholic Fatty Liver Disease

Non-alcoholic fatty liver disease (NAFLD) is a metabolic liver condition characterized by the presence of hepatic steatosis in the absence of alcohol consumption in toxic quantities. NAFLD is related to MetS and its components, such as obesity, glucose intolerance and hypertriglyceridemia. The mechanisms underlying NAFLD have yet to be fully elucidated, although there are suggestions that IR, concomitant hyperinsulinemia, increased flow of FFAs to the liver and elevated inflammatory mediators are involved in the pathogenesis of this condition, in parallel with the development of MetS. Regardless of age, gender and body mass index (BMI), studies have shown that people with MetS had a considerably higher liver fat content than those without the disease. The pathophysiology and IR that both disorders share, along with the fact that obesity is the root cause of both, serve as connecting elements for NAFLD and MetS [26,27].

## 4. Possible Biomarkers of Metabolic Syndrome

### 4.1. Lipid Profile

Dyslipidemia is an intrinsic condition of MetS, characterized by the presence of high TG levels, low HDL-C and high LDL-C. This dyslipidemia is strongly associated with CVD and T2DM in MetS patients. IR and central obesity, both components of MetS, are correlated with various abnormalities in the lipid profile. The IR and secondary hyperinsulinemia observed in MetS promote the overproduction of LDL particles. In addition, a relative deficiency of lipoprotein lipase results in decreased removal of TG-rich lipoproteins (TRLs) during fasting periods and after meals, as well as reduced production of HDL particles. The resulting rise in the concentration of ester-rich TRLs during the fasting and postprandial periods is a central feature of MetS-associated dyslipidemia [28].

IR or the absence of insulin stimulates lipolysis in intra-abdominal fat, a metabolically active region that releases FFAs. In the liver, these FFAs are converted into TG, contributing to hypertriglyceridemia, a common component of MetS and an important risk factor for CVD. In addition, high LDL-C levels are also a significant risk factor for CVD, and reducing these levels is a priority in pharmacotherapy aimed at cardiovascular prevention.

An association between increased TG levels and decreased LDL-C with MetS and CVD has been proposed [29]. Recent studies such as that by Nur Zati Iwani et al. and Nie et al. have shown that the TG/HDL-C ratio is a convenient tool for detecting IR, one of the main components of MetS, Table 1 [30,31].

Apolipoproteins are proteins synthesized in the liver that are essential for the transport and redistribution of lipids. Apolipoprotein A1 (ApoA1), present in HDL, facilitates the reverse transport of cholesterol from peripheral tissues to the liver, exerting an antiatherogenic effect. In contrast, apolipoprotein B (ApoB) is involved in the transport of cholesterol from liver cells to peripheral cells, increasing the risk of atherosclerosis and CVD as it promotes the deposition of cholesterol in the arteries. Thus, apolipoproteins play a crucial role in the transport of lipids in the bloodstream and are recognized as important markers of lipid profile and cardiovascular risk. While traditional cholesterol measurements provide an overview of lipid transport, serum apolipoprotein levels offer a more direct assessment of the number and composition of circulating lipoprotein particles. Specifically, Apo B levels reflect the amount of TG-rich very-low-density lipoprotein (VLDL) particles and the number of LDL particles. Similarly, ApoA1 levels correspond to the amount of HDL [44].

A study by Kir et al. showed evidence of high ApoB levels, low ApoA1 levels and a high ApoB/ApoA1 ratio associated with MetS. It should be emphasized that the ApoB/ApoA1 ratio is considered a more accurate indicator for assessing the balance between atherogenic and antiatherogenic lipoproteins. According to this study, ApoB/ApoA1 predicts cardiovascular risk more accurately than any of the cholesterol indices, and this ratio also increased significantly with the increase in the number of MetS components, Table 1 [32]. Thus, this ratio is a more effective predictor of cardiovascular risk related to cardiovascular lipoproteins than traditional lipid markers. It is suggested that this ratio would be an ideal marker for lipid alterations associated with IR and MetS since it covers the main characteristics of dyslipidemia linked to IR and MetS, including low levels of LDL-C and high levels of VLDL and LDL-C [32,44].

### 4.2. Inflammatory Markers

#### 4.2.1. Interleukin 6

Recent studies such as that by Kir et al. have shown a significant correlation between IL-6 and the five components of MetS, Table 1 [32]. This association is largely explained by the dysfunction of adipose tissue, which stimulates the proliferation of macrophages and consequently increases the production of IL-6. It is suggested that the presence of high levels of IL-6 in hepatocytes during states of chronic inflammation may play a key role in the development of MetS. This finding indicates a possible mechanism by which IL-6 contributes to MetS physiopathology, adding further insight into the role of this cytokine in the complexity of this condition [7].

#### 4.2.2. Tumor Necrosis Factor Alpha

TNF-α, like IL-6, is secreted by inflammatory cells present in dysfunctional adipose tissue. This cytokine plays a central role in various pathways and dysfunctions associated with MetS, including IR, through mechanisms involving the activation of the mammalian target of rapamycin (mTOR) and protein kinase C (PKC), as well as systemic inflammation [34]. Chronic inflammation induced by TNF-α can lead to the activation of the mTOR pathway in different cells and tissues of the body and may contribute to the metabolic dysfunction associated with MetS. mTOR is a component of a complex signaling pathway involved in the abundance of energy, nutrients and metabolism. The mTOR signaling pathway is implicated in several pathological conditions, including T2DM and obesity. Some reports suggest that rapamycin has critical effects on pancreatic β-cells, negatively affecting glucose-stimulated insulin secretion in β-cells. mTOR signaling is overactivated in obesity, promoting inflammation and IR. In this sense, there is great interest in developing mTOR inhibitors as therapeutic drugs for obesity and diabetes [45].

Obesity induces a systemic inflammatory state that results in dysfunctions in macrophages and adipocytes, leading to inappropriate production of cytokines, including TNF-α. High levels of TNF-α contribute to IR through various mechanisms, thus promoting the progression of MetS. Studies by Monserrat-Mesquida et al. and Tylutka et al. showed that TNF-α levels are higher in individuals with MetS compared to individuals without MetS, Table 1 [33,34]. Thus, TNF-α plays a crucial role in the pathogenesis and progression of MetS and may be a useful marker in identifying patients with this condition.

#### 4.2.3. Plasminogen Activator Inhibitor Type 1

PAI-1, which belongs to the serine protease inhibitor family, plays a critical role in regulating the fibrinolytic system, inhibiting tissue-type plasminogen activator (tPA) and urokinase-type plasminogen activator (uPA), which are unable to convert plasminogen into plasmin. When PAI-1 inhibits these activators, it triggers a prothrombotic state and contributes to the development of cardiovascular events [14].

As the main inhibitor of fibrinolysis, high levels of PAI-1 can increase the risk of coronary heart disease. Increased PAI-1 is involved in controlling insulin signaling in adipocytes and can be considered a component of MetS [36].

Under physiological conditions, PAI-1 is secreted into the circulation or extracellular space by endothelial cells, adipocytes, vascular smooth muscle cells, platelets or hepatocytes. However, under pathological conditions, PAI-1 is induced by many pro-inflammatory and pro-oxidant factors. For example, when TNF-α, angiotensin II, glucocorticoids and insulin are elevated, adipocytes are stimulated to increase PAI-1 levels. Hypoxia and Reactive Oxygen Species (ROS) also increase PAI-1 levels. High levels of PAI-1 consequently affect the vasculature, inflammatory signaling, adiposity and IR [46].

The association between PAI-1 and MetS has been firmly established over time, showing a significant correlation between high PAI-1 levels and MetS severity. Nawaz et al. demonstrated a strong association between PAI-1 and the constituent elements of MetS, such as BMI, TG and IR, Table 1 [35]. Given that PAI-1 levels are consistently high in individuals with MetS, PAI-1 plays a direct role in the regulation of lipid metabolism, contributing to cardiovascular complications in MetS and in the molecular pathogenesis of MetS itself. They also predict that new therapeutic agents that reduce the synthesis or block the function of PAI-1 will be valuable in the prevention and treatment of MetS and associated vascular consequences [47].

#### 4.2.4. C-Reactive Protein

CRP is a non-specific biomarker commonly used in the assessment of disease activity, differential diagnosis or classification of inflammatory diseases [48]. It is an acute phase reagent and is produced by hepatocytes, this production being regulated by IL-6 and other inflammatory cytokines [49]. This pro-inflammatory cytokine has been associated with obesity, MetS and other MetS-related disorders and predicts the risk of acquiring IR and T2DM [50]. Cura-Esquivel et al. showed that serum CRP levels were higher in obese individuals with MetS, Table 1 [36].

#### 4.2.5. Leptin

Leptin is an adipokine which, under normal physiological conditions, functions to reduce appetite, increase energy expenditure, increase sympathetic activity, facilitate glucose utilization and improve insulin sensitivity. It is expressed at levels proportional to fat mass, and although it is mainly produced by adipocytes, it is also produced by vascular smooth muscle cells, cardiomyocytes and the placenta in pregnant women. Implications for leptin cover a wide range of physiological and pathological processes, including regulation of energy homeostasis, obesity syndromes, metabolic dysfunctions, neuroendocrine function and bone metabolism [51].

In the hypothalamus, leptin binds to OB-R surface receptors, triggering the activation of signaling pathways, including the Janus kinase/signal transducers and activators of transcription (JAK/STAT) pathway.

This signaling has specific effects on the expression of peptides that regulate appetite. More specifically, activation of the JAK/STAT pathway by leptin increases the expression of anorexigenic peptides, such as pro-opiomelanocortin (POMC) and cocaine and amphetamine gene-related peptide (CART), which promote satiety and reduce appetite. Simultaneously, leptin inhibits the expression of orexigenic neuropeptides, such as neuropeptide Y (NPY) and agouti-related protein (AgRP), which stimulate appetite. These coordinated mechanisms contribute to increased energy expenditure and reduced appetite, playing an important role in regulating energy balance and maintaining metabolic homeostasis [52].

It has been proven that leptin concentrations are significantly increased in obesity [53,54,55]. High leptin levels lead to global and/or selective leptin resistance, a condition prevalent in individuals with obesity and MetS. Inadequate leptin levels can trigger the development of T2DM and CVD. MetS is a condition that favors leptin resistance through systemic inflammation, IR, hyperlipidemia, hypertension, atherosclerosis and obesity [17,56].

#### 4.2.6. Adiponectin

Adiponectin is considered a protective adipokine since it is antiatherogenic, anti-inflammatory and antidiabetic [57]. In terms of circulating levels, it has been observed that adiponectin is more abundant in women than in men, a phenomenon attributed to the stimulating influence of testosterone on its secretion. This hormone plays a crucial role in metabolic balance, and reduced levels are associated with increased cardiovascular, vascular and metabolic risk [51].

It serves as a critical messenger to mediate cross-communication between adipose tissue and other metabolic tissues and organs. Functionally, adiponectin binds to the surface receptors adipoR1 and adipoR2 and activates downstream signaling pathways, such as the AMP-activated protein kinase (AMPK) pathway, to increase glucose uptake and fatty acid oxidation [52]. In healthy individuals, adiponectin plays essential physiological roles in regulating metabolism. However, in individuals with MetS, adiponectin levels are often decreased. Studies have highlighted the positive impact of this hormone on metabolic protection, especially due to its potential inhibitory activity in the atherogenic process [13].

There is an inverse correlation between adiponectin levels and MetS components. Hypoadiponectinemia associated with visceral obesity is directly related to vascular alterations and IR. These findings highlight the importance of adiponectin as a significant marker in the assessment of MetS and associated complications, as well as emphasizing its potential as a therapeutic target in metabolic interventions [51].

#### 4.2.7. Leptin/Adiponectin Ratio

Several studies have pointed to the potential of the leptin/adiponectin ratio as a valuable indicator due to its ability to improve the prediction of cardiovascular and metabolic events. Recent evidence suggests that these two molecules exert antagonistic effects, which reinforces the interest of leptin/adiponectin as a marker of metabolic dysfunction [13,37,51]. A study by Lee et al. showed that higher levels of leptin/adiponectin are associated with a greater risk of developing MetS. They also determined that the leptin/adiponectin ratio is a more accurate marker than adiponectin or leptin alone for identifying individuals at risk of MetS, Table 1 [37].

#### 4.2.8. Chimerin

Chimerin is an adipokine that has recently gained attention as a potential MetS biomarker given its involvement in inflammation, glucose metabolism, adipogenesis, angiogenesis and whole-body energy homeostasis [7].

It is highly expressed in white adipose tissue, liver and lungs. It binds to the surface receptors CMKLR1 and GPR1 and activates downstream signaling pathways, including the MAPK, ERK1/2, and PI3K-Akt pathways to regulate adipogenesis, angiogenesis and inflammation. In addition, chimerin also regulates glucose metabolism through actions on insulin secretion and suppression of hepatic glucose production [52]. A study by Ba et al. revealed a significant increase in circulating levels of chimerin in children and adolescents who met the criteria for MetS [58]. However, Pelczyńka et al. showed no differences in chimerin concentrations between the MetS and non-MetS groups. Therefore, more studies are needed, as there are studies that do not confirm the relationship between chimerin and MetS, and thus, the association of serum chimerin levels with MetS cannot be clearly established [59].

#### 4.2.9. Omentin

Omentin (OMEN) is a protein present in two forms, OMEN-1 and OMEN-2, OMEN-1 being the main isoform in human plasma. Unlike most adipokines, OMEN is not produced in subcutaneous adipose tissue but is mainly synthesized in visceral adipose tissue [60].

OMEN-1 has been shown to exert anti-inflammatory effects, and the expression of the omentin gene and serum levels of omentin are lower in obese people, correlating negatively with measures such as waist circumference, BMI, IR and glucose intolerance. On the other hand, there is a positive correlation between omentin, serum adiponectin and HDL-C and a negative correlation with leptin levels [60]. Sun et al. revealed that a decrease in the level of omentin in adolescents is an independent predictor of MetS and obesity, and of all the components of MetS, it is most closely associated with central obesity. They suggested that omentin, as a link between MetS, obesity and CVD, could become a simple and effective serological marker for predicting and monitoring the intervention of these conditions, Table 1 [38].

OMEN-1 promotes vasodilation by increasing nitric oxide (NO) synthesis and reducing TNF-α production. Low serum concentrations of OMEN have been observed in patients with type 1 and type 2 diabetes, as well as being associated with lower levels in patients with coronary artery disease. Omentin increases insulin-induced glucose uptake and participates in the regulation of insulin sensitivity and may exert a protective action against worsening IR [60].

Decreased OMEN-1 levels have also been correlated with a series of cardiovascular complications, including heart failure and adverse cardiac events [61]. Therefore, serum OMEN-1 levels emerge as a potential biomarker for vascular dysfunction, atherosclerosis and cardiovascular risk in various conditions, such as coronary artery disease, obesity, inflammatory disease, T2DM and MetS. In addition, omentin could become a future pharmacotherapeutic target that should be further investigated through new studies.

#### 4.2.10. Fetuin-A

Fetuin-A is a 64 kDa glycoprotein predominantly secreted by the liver and adipose tissue. Its presence contributes to the migration of macrophages into adipose tissue and increases the expression of pro-inflammatory cytokines such as IL-6 and TNF-α while reducing the expression of adiponectin [14]. According to Pan et al., the concentration of circulating fetuin-A in MetS patients was significantly higher than in control groups, with a slight tendency for the risk of MetS to increase with an increase in the concentration of circulating fetuin-A [62].

Furthermore, given that there is a significant positive correlation between fetuin-A and MetS and a significant negative correlation between adiponectin and MetS, a study by Zhou et al. showed that the fetuin-A/adiponectin ratio is a more sensitive indicator for assessing MetS than fetuin-A and adiponectin alone, Table 1 [39].

### 4.3. Oxidative Stress

OS biomarkers include molecules modified by redox stress in the microenvironment and components of the antioxidant system that are altered in response to OS. Risk factors interrupt cell signaling pathways, promoting an increase in inflammatory markers, lipid peroxides and free radicals, resulting in cell damage and clinical manifestations of MetS. It is postulated that OS and inflammatory markers contribute to the pathogenesis of MetS. The quantification of biomarkers represents the most accurate method for assessing the presence of OS in vivo. In addition, the assessment of total antioxidant capacity can be used as a measure of OS in MetS. Isoprostanes (IsoP), derivatives of arachidonic acid, especially 8-iso-prostaglandin-F2α, can be a valuable tool for simultaneously investigating OS and inflammation in conditions where both are implicated [63,64].

Studies have been conducted in individuals with MetS in which the concentrations of OS biomarkers and antioxidant enzyme activity were assessed simultaneously. The findings indicate that the presence of MetS is associated with an increase in OS biomarkers and a reduction in antioxidant capacity, evidencing a pro-inflammatory condition and compromised health as part of a complex process that contributes to cardiometabolic diseases [64].

OS biomarkers are fundamental in studies aimed at identifying patients at risk of complications and targeting appropriate therapy to mitigate the burden of MetS. These markers include substances that reflect lipid peroxidation and protein and amino acid oxidation, as well as DNA oxidation. Among them, thiobarbituric acid (TBARS), malondialdehyde (MDA), 4-hydroxy-2-nonenal (4-HNE) and F2-isoprostanes stand out as indicators of lipid peroxidation, while protein carbonyls, advanced glycation products (AGEs), oxidized LDL (ox-LDL) and oxidized proteins represent protein oxidation. For DNA oxidation, markers include 8-oxo-2′-deoxyguanosine (8-oxo-dG), 5-chlorouracil and 5-chlorocytosine [65].

A common feature of MetS-associated dyslipidemia is the elevation of small, dense LDL particles, which are easily oxidized. Hurtado-Roca et al. demonstrated that high levels of ox-LDL are associated with IR, which is closely linked to the pathogenesis of MetS. IR may result from OS-mediated activation of kinase signaling cascades that phosphorylate insulin receptors, leading to impaired insulin action. It is also related to the correlation between plasma glucose and the susceptibility of LDL to oxidation, since high concentrations of glucose can induce LDL oxidation. In addition, obesity is the main origin of MetS and has been implicated in the induction of OS, which in turn may contribute to the development of MetS. Ox-LDL levels may reflect the central mechanisms through which MetS components develop and progress in parallel with IR and may be an early sign of MetS development, Table 1 [40].

Markers associated with ROS generation include xanthine oxidase, gamma-glutamyltransferase (GGT), myeloperoxidase (MPO), NADPH oxidase and nitric oxide synthase (NOS). GGT, an enzyme widely distributed in the body, is involved in the metabolism of glutathione (GSH), an antioxidant and metabolic substrate. High levels of GGT have been associated with MetS, diabetes, hypertension and the risk of stroke. In addition to enzymatic markers, there are also non-enzymatic markers such as glycoprotein A (GPA), CRP, ferritin and uric acid. Serum ferritin, which is positively correlated with liver damage and IR, plays an important role in the diagnosis of MetS [17].

Although uric acid can act as a direct antioxidant in some situations, neutralizing free radicals and reducing OS, it is important to consider that in MetS, high levels of uric acid are often associated with conditions that promote OS and chronic inflammation. In a hydrophobic environment, uric acid loses its antioxidant capacity, and in the presence of lipid peroxides, it even becomes a strong pro-oxidant. Thus, the intracellular environment of adipocytes, which is predominantly hydrophobic and has a high level of ROS, is largely unfavorable for the manifestation of uric acid’s antioxidant properties [66].

During the process of uric acid formation by the action of xanthine oxidases, there is simultaneous production of ROS. Both intracellular uric acid and ROS influence various intracellular signaling pathways, alterations in which can contribute to the development of atherosclerotic lesions [66,67]. Studies such as that by Jeong et al. have shown that increased uric acid levels are positively correlated with obesity, MetS, T2DM and CVD, Table 1 [41,48]. It is suggested that hyperuricemia may be one of the causal factors that induce OS followed by a pro-inflammatory process and endocrine dysfunction in adipose tissue, contributing to the pathogenesis of MetS and CVD [66].

### 4.4. MiRNAs

MiRNAs have emerged as key regulators in a variety of biological processes and play a crucial role in metabolic diseases. The role of miRNAs as key regulators of metabolic homeostasis has been intensively explored in the last decade. Brandao et al. grouped the significant circulating miRNAs that are altered in obese individuals, where miRNAs such as miR-92a-3p, miR-151a and miR-155 were shown to be increased. On the other hand, miR-26a, miR-30b, miR-30c, miR-125b, miR-126, miR-139-5p, miR-144-5p, miR-223 and miR-376a are reported to be decreased in obese adults when compared to healthy lean individuals [68].

Studies have suggested that certain miRNAs, such as miR-24, may play an important role in the regulation of lipid homeostasis and in the development of metabolic diseases such as NAFLD and obesity. High levels of miR-24-3p expression have been found in obese children with MetS, highlighting its role in the pathogenesis of childhood obesity and MetS. Zhang et al. suggested that miR-24-3p may play a significant role in the pathogenesis of childhood obesity and may be a promising marker for predicting MetS in obese children, Table 1 [42].

A study by Ramzan et al. identified miR-15a-5p and miR5-17-5p as important predictors in the presence of MetS. These miRNAs share genes involved in the regulation of metabolic pathways, including insulin, fatty acid metabolism and AMPK, Table 1 [43].

In addition to these, five overexpressed molecules (miR-142-3p, miR-140-5p, miR-222 miR-143 and miR-130) and five underexpressed molecules (miR-532-5p, miR-423-5p, miR-520c-3p, miR-146a and miR-15a) were identified, all positively linked to adipokine concentrations. The molecules that showed positive regulation correlated positively with BMI, diabetes and lipid profile parameters, while those that were negatively regulated showed negative correlations with these same parameters. MiR-122, described as being involved in hepatic steatosis, and miR-192 correlated with TNF-α and procalcitonin and negatively with adiponectin. Similar correlations were observed for miR-34a and TNF-α and procalcitonin. This suggests that these three miRNAs may play a role in the obesity-related inflammatory reaction, which is an important feature of MetS pathogenesis [69].

Studies have illustrated a correlation between the expression of miRNAs in adipose tissue and different metabolic parameters, such as BMI, adipogenesis, glycemia and leptinemia [70]. Adipokines modulate the expression of miRNAs involved in adipogenesis [70]. For example, miR-218 can interact with the 3′UTR region of the mRNA encoding the adiponectin receptor, resulting in inhibition of the effect of adiponectin on glucose uptake.

In addition, they revealed that miRNAs can modulate the expression and/or secretion of TNF-α. They observed that miR-145 stimulates TNF-α expression in adipocytes through activation of the NF-κB pathway, leading to increased expression of miR-130, miR-146b, miR-150, miR-221 and miR-222 and decreased levels of miR-103 and miR-143 [71].

These results highlight a wide range of miRNAs that influence the pathophysiology of MetS. They have shown potential as accurate and sensitive biomarkers for MetS, reflecting the underlying molecular changes associated with this condition. Detection in biological fluids offers a particular advantage, allowing early and non-invasive diagnosis as well as a better understanding of the progression of MetS over time.

## 5. Macroalgae

In recent decades, various natural compounds have been shown to exert multiple beneficial effects, including anti-inflammatory, antihyperglycemic and antilipemic activities, which contribute to improving lipid and blood glucose levels as well as increasing insulin sensitivity. Among these natural compounds, seaweed has received increasing attention due to its rich micro- and macronutrient profile. These products have been and continue to be used in nutraceuticals and functional foods for the management of comorbidities associated with MetS [72].

Seaweed has a remarkable nutritional profile, characterized by a high content of fiber, minerals and polyunsaturated fatty acids. In addition to these nutrients, seaweed is rich in several unique bioactive compounds, such as phlorotannins and polysaccharides, which are not found in terrestrial plants. These bioactive compounds have the potential to modulate the progression of chronic diseases [73].

Seaweed are an abundant source of dietary fiber, which is essential for digestive health. In addition, they contain a variety of essential minerals, such as iodine, iron, calcium and magnesium, which play vital roles in various biological functions [73]. The polyunsaturated fatty acids present in high concentrations in seaweeds are known for their anti-inflammatory and cardioprotective properties. Unique compounds in seaweeds, such as phlorotannins, have antioxidant and anti-inflammatory activities, while polysaccharides can positively influence the immune response and intestinal health [72,74].

Epidemiological studies have shown a correlation between regular consumption of seaweed and a reduction in the prevalence of various chronic diseases [75]. Dietary intake of seaweeds is associated with a lower risk of developing CVD and hyperlipidemia due to the presence of compounds that help regulate lipid levels and protect against oxidative damage [76].

In addition, bioactive compounds in seaweed, such as polyphenols, have demonstrated anticarcinogenic properties, suggesting a potential role in cancer prevention. Bioactive molecules present in seaweed, such as polyunsaturated fatty acids and polyphenolic compounds, have properties that can help prevent and manage T2DM by improving insulin sensitivity and regulating blood glucose [72].

The consumption of seaweed offers not only a rich source of nutrients but also a variety of bioactive compounds that can contribute significantly to the prevention and management of chronic diseases, making them a valuable functional food [76].

Seaweeds include more than 25,000 species and are fundamental to marine ecosystems. They act as important blue carbon sinks, essential for a sustainable economy. Their diversity, abundance and rapid proliferation make them promising biological resources for the future. Seaweeds are classified into three main groups based on the color of their pigments: brown (Phaeophyceae), red (Rhodophyta) and green (Chlorophyta) [72].

### 5.1. Macroalgae Nutritional Compounds

#### 5.1.1. Lipids and Fatty Acids

The lipids and fatty acids in macroalgae play essential roles as components of cell membranes, energy reserves and cell signaling molecules. Although marine macroalgae generally contain low levels of lipids (2–4.5% of dry weight), these lipids are rich in long-chain polyunsaturated fatty acids (PUFAs), such as eicosapentaenoic (EPA) and docosahexaenoic (DHA) acids. These PUFAs, which belong to the omega-3 and omega-6 families, are essential for cardiovascular and neurological health and for modulating inflammation [72,77]. Red and brown macroalgae are particularly rich in EPA, which can account for up to 50% of the total fatty acid content in some species, such as *Palmaria palmata* (Rhodophyta) (Figure 1a) [77]. Long-chain PUFAs, especially EPA and DHA, are recognized for their benefits in reducing triglycerides, improving endothelial function and having anti-inflammatory effects, and they are important components in functional foods and nutraceuticals [78].

Phycobiliproteins are used commercially for their therapeutic value as anti-inflammatory agents. R-phycoerythrin, found in various red algae, such as *Gelidium pusillum*, *Chondrus crispus* (Figure 1b) and *Gracilariopsis longissima* (formerly *Gracilaria verrucosa*) (Figure 1c) (Rhodophyta), has demonstrated anti-inflammatory properties. Phycocyanin also showed anti-inflammatory activity, accompanied by a remarkable antioxidant capacity [78].

In addition to PUFAs, macroalgae contain sterols, such as fucosterol, which have potential health benefits, including cholesterol regulation and cardiovascular protection [77]. These lipid compounds are increasingly valued not only for their nutritional function but also for their therapeutic potential in various health conditions [76,77].

#### 5.1.2. Proteins

The proteins found in macroalgae, such as lectins and phycobiliproteins, are crucial components with multiple biological functions. Lectins, for example, are glycoproteins that bind specifically to carbohydrates, playing important roles in intercellular communication, modulation of the immune system and protection against pathogens [72].

Phycobiliproteins, on the other hand, are fluorescent pigments responsible for capturing light in red algae and are used in biotechnological applications, such as fluorescent probes in molecular biology. These proteins also exhibit antiviral, antibacterial and anticancer activities, making them of interest in research into new therapies [72].

In addition, seaweed proteins are a rich source of essential amino acids, such as glycine, arginine and glutamic acid, which are fundamental for protein synthesis in the human body and for the proper functioning of the immune system [72,77].

#### 5.1.3. Polysaccharides

In marine algae, polysaccharides are biomacromolecules made up of repeating monosaccharide units joined by glycosidic bonds. The taxonomic distribution of polysaccharides varies according to the type of algae: in brown algae (Ochrophyta, Phaeophyceae), fucoidans, alginates and laminarins predominate; in red algae (Rhodophyta), porphyran, carrageenan and floridean starch are the main ones; and in green algae (Chlorophyta), rhamnan sulphate and ulvan stand out [72,73]. These compounds are known for their anticarcinogenic, antioxidant, anti-inflammatory and antidiabetic properties. For example, fucoidans are sulphated polysaccharides that have shown anticancer activities and positive effects on modulating the intestinal microbiota, which is crucial for preventing metabolic diseases [73].

Red algae contain agar and carrageenan, which are polysaccharides widely used in the food and pharmaceutical industries. Agar is a polysaccharide sulphate found in the cell walls of red macroalgae such as *Gelidium* and *Gracilaria* (Rhodophyta). Carrageenan extracted from *Chondrus crispus* (Rhodophyta) is a sulphate polysaccharide, also present in the cell walls of red macroalgae. These compounds have gelling and bioactive properties, such as immunomodulatory, antioxidant and anticancer activities. They have been shown to inhibit digestive enzymes such as α-amylase and α-glucosidase, resulting in a reduction in blood glucose levels [72].

Green algae contain rhamnan sulphate and ulvan, both sulphated polysaccharides that have antioxidant, anti-inflammatory and anticancer activities [73,77].

Thus, polysaccharides extracted from macroalgae have antioxidant activity, contributing to the neutralization of reactive ROS, whose production is intensified by hyperglycemia associated with diabetes, leading to OS and inflammation [79].

#### 5.1.4. Peptides

The bioactive peptides present in macroalgae are composed of amino acid sequences which, when released through digestion, food processing or fermentation, exhibit significant biological activities. These peptides, which can range from 3 to 40 amino acids, are known for their antioxidant, antihypertensive, anticarcinogenic and antiatherosclerotic properties [72].

For example, peptides derived from red and brown algae have shown potential in regulating blood pressure by promoting vasodilation through the inhibition of angiotensin-converting enzyme (ACE), a common mechanism in hypertension treatments. In addition, peptides from algae have shown antitumor activities, interfering with cell proliferation and inducing apoptosis in cancer cells. Their antioxidant capacity is also remarkable, neutralizing free radicals and protecting cells against OS, a key factor in the development of various chronic diseases [73].

#### 5.1.5. Polyphenols

Polyphenols are bioactive compounds widely studied for their potent antioxidant properties. In macroalgae, the most common polyphenols include phlorotannins, bromophenols and flavonoids, each with distinct chemical structures and varied biological activities [4].

Phlorotannins, found mainly in brown algae, are known for their neuroprotective, antidiabetic, anti-inflammatory and anticancer activities. Phlorotannins isolated from the brown alga *Ecklonia cava*, one of the most abundant sources of polyphenolic compounds, have shown antioxidant properties. These compounds can inhibit the formation of free radicals and protect against oxidative damage, which makes them valuable in the prevention of neurodegenerative and CVD [4].

The bromophenols present in red algae show a wide range of biological activities, including antioxidant, antimicrobial, antidiabetic and antiobesity properties, making them potential candidates for the development of new therapeutic agents. Bromophenols isolated from *Symphyocladia latiuscula* (Rhodophyta) have shown significant antioxidant activity, characterized by their structure with multiple highly brominated groups, especially 3,4-dihydroxy-2,5,6-tribromobenzyloxy. Similarly, *Polysiphonia stricta* (Rhodophyta) also showed antioxidant activity, suggesting that the antioxidant potential of red algae is related to the presence of brominated units and the degree of bromination of the molecules [73,80].

Flavonoids are phenolic compounds structurally composed of a heterocyclic oxygen ring linked to two aromatic rings, the variations of which occur according to the degree of hydrogenation. These compounds are widely distributed in terrestrial plants and algae, with more than 2000 variants identified, classified into categories such as flavones, flavonols, flavanones, anthocyanins and isoflavones. Flavonoids such as rutin, quercetin and hesperidin have been detected in various species of Chlorophyta, Rhodophyta and Phaeophyceae. Isoflavones such as daidzein and genistein are present in red macroalgae such as *Chondrus crispus* and *Porphyra/Pyropia* spp. (Rhodophyta) and in brown algae such as *Sargassum muticum* (Figure 2a) and *Sargassum vulgare* (Phaeophyceae). Flavonoid glycosides have also been found in brown algae, including *Durvillaea antarctica*, *Lessonia spicata* and *Macrocystis pyrifera* (Phaeophyceae) [4,80].

These flavonoids are closely associated with various biological properties, including anti-inflammatory, anticancer and cell proliferation-suppressing effects, contributing to antiatherosclerotic and antihypertensive effects. In addition, flavonoids exert an antiaggregant action, neutralizing platelet hyperactivity, which is a risk factor in CVD. They also have beneficial metabolic effects on obesity and atherosclerosis, extending their therapeutic potential in the treatment of CVD [4,78].

#### 5.1.6. Phytochemicals

Phytochemicals, such as carotenoids and polyunsaturated fatty acids (PUFAs), are non-nutritional but essential compounds that play vital roles in maintaining human health. Carotenoids, such as fucoxanthin, β-carotene, lutein and astaxanthin, are terpenoid pigments that provide protection against damage caused by UV light and have significant antioxidant properties [75,77].

Fucoxanthin, found mainly in brown algae, has attracted attention for its anticancer properties, particularly in inducing apoptosis in tumor cells and inhibiting tumor growth. β-carotene, a precursor of vitamin A, is essential for eye health and has strong antioxidant properties [72]. Fucoxanthin, extracted from *Undaria pinnatifida* (Phaeophyceae) (Figure 2b), and its metabolite fucoxanthinol have been shown to inhibit the differentiation of adipocytes and the accumulation of lipids in the liver. These compounds act by reducing the expression of adipogenic and lipogenic factors while promoting the upregulation of the thermogenic mitochondrial protein UCP-1 in white adipose tissue, resulting in an antiobesity effect [81].

Polyunsaturated fatty acids (PUFAs), especially omega-3 and omega-6 fatty acids, are important components in seaweed [72]. These compounds are essential for cardiovascular health, brain function and regulating inflammatory responses. ACEAs such as eicosapentaenoic acid (EPA) and docosahexaenoic acid (DHA) are recognized for their ability to reduce the risk of CVD, improve cognitive function and protect against neurodegenerative diseases [72,75].

The inclusion of bioactive components derived from macroalgae in the diet could be a promising strategy for the management and prevention of MetS. Their ability to modulate the intestinal microbiota, reduce inflammation, improve insulin sensitivity and regulate the lipid profile makes these compounds potential candidates for dietary interventions focused on metabolic diseases [72,81]. Future studies should continue to explore the efficacy of these compounds in human clinical trials, with a view to developing new functional food-based therapies for MetS.

## 6. Seaweed-Based Dietary Interventions for Metabolic Syndrome

Changes in lipid profile represent one of the main risk factors for the development of CVD, which is recognized as the leading cause of global mortality. In response to the need to reduce mortality rates, epidemiological studies indicate that the consumption of seaweed can induce antilipemic effects. The intake of fiber-rich foods has been widely described as a protective factor against various pathologies, such as hypertension, obesity and diabetes. Algae, being rich sources of soluble fibers such as carrageenans, play a relevant role as a source of dietary fiber, contributing to the prevention of these conditions [82].

Carrageenans, polysaccharides extracted from red seaweed, have been the subject of study due to their potential impact on MetS [83]. Scientific studies have shown that carrageenans have the ability to reduce total cholesterol (TC) and LDL-C levels, especially in women, when consumed regularly [82]. A study analyzed the effects of consuming carrageenan in a vegetable jelly on the lipid profile of 30 individuals with hypercholesterolemia, aged between 20 and 64, for 60 days. The results showed a significant reduction of 5.3% in TC levels after the consumption period, with this reduction being more pronounced in women. In addition, LDL-C levels also showed a significant drop of 5.4% in women. However, there was an increase in TG levels after 60 days, especially among men [82]. This beneficial effect is associated with the ability of carrageenans to capture cholesterol and bile salts, inhibiting the action of lipase and consequently decreasing the absorption of cholesterol in the intestine. However, it is important to consider that the consumption of carrageenans can also lead to an increase in TG levels. This increase may be related to the ability of carrageenans to increase the viscosity of intestinal contents, leading to a prolonged digestion time and reducing the efficiency of digestion. In addition, carrageenans can inhibit the action of lipase, which can result in a decrease in micelle formation, indirectly affecting cholesterol absorption. Therefore, although carrageenans have shown beneficial effects in lowering cholesterol, especially LDL-C, it is essential to consider the possible adverse effects, such as increased TG levels, when assessing their impact on MetS [82]. More studies are needed to fully understand the role of carrageenans in metabolic and cardiovascular health.

Alginates, polysaccharides derived from brown seaweed, have multiple beneficial effects in the management of MetS, acting on various risk factors such as glucose control, cholesterol reduction and improved intestinal health [84]. As an important source of soluble fiber, alginates are effective in reducing cholesterol and controlling glucose levels. Fiber intake has been widely associated with reducing the risk of CVD and controlling diabetes [21].

When ingested, alginates form a viscous gel in the gastrointestinal tract, which slows down the absorption of nutrients such as fats and carbohydrates. This process results in a decrease in glucose and insulin spikes after meals, contributing to glycemic control and helping to regulate blood glucose levels [84,85].

*Sargassum fusiforme* (Phaeophyceae) alginate influences the intestinal microbiota, which has a positive impact on metabolic and inflammatory regulation. Changes in the composition of the intestinal microbiota are linked to the development of MetS, and the consumption of alginates can help improve the composition and function of this microbiota, favoring metabolic balance [86].

Alginates promote a greater feeling of satiety by delaying gastric emptying, helping to reduce calorie intake. In addition, they can inhibit the absorption of fats, decreasing lipid digestion and the absorption of TG, aiding weight loss and obesity control [84]. A randomized, double-blind, placebo-controlled intervention study in 50 Japanese subjects with a BMI ≥ 25 and <30 kg/m^2^ was conducted to investigate whether supplementation with iodine-reduced kelp (*Saccharina japonica*) powder decreases body fat composition in overweight Japanese subjects. Participants were supplemented with iodine-reduced kelp powder (3 g alginate/day) for 8 weeks. The results showed a significant reduction in the percentage of body fat among the men who consumed the supplement, suggesting that alginate may be effective in losing body fat. However, this reduction was not observed in women. In addition, LDL-C levels decreased in the non-hyperlipidemic participants who consumed kelp, suggesting a beneficial effect of alginate in lowering cholesterol. There was no increase in thyroid hormone levels, indicating that alginate supplementation, even with additional iodine intake, is safe for individuals with high iodine intake from algae [87]. In addition, alginates are widely used as drug coatings, offering sustained and prolonged release, which is particularly useful for continuous treatments, such as the use of hypoglycemic drugs in MetS patients [84].

Dietary carotenoids play an important role in controlling hypertension and may be beneficial for MetS. Carotenoids, such as astaxanthin (ASX), β-carotene, lycopene and lutein, act as potent antioxidants, fighting OS [88]. A randomized, placebo-controlled clinical study demonstrated that oral administration of 8 mg/day of ASX for 8 weeks in patients with T2DM resulted in an increase in serum adiponectin concentration and a reduction in visceral body fat mass. The results also indicated decreases in TG levels, LDL-C and systolic blood pressure (SBP). There was also a significant reduction in the concentration of fructosamine and a marginal decrease in plasma glucose [89].

ASX has a wide range of biological activities, such as antioxidant, anti-inflammatory, antidiabetic, immunomodulatory and antihyperlipidemic effects. Due to its potent antioxidant capacity, ASX is 5 to 15 times more effective than other carotenoids in eliminating free radicals, which boosts its antihypertensive properties [88].

Studies have shown that ASX supplementation in hypertensive animal models significantly reduced SBP and diastolic blood pressure (DBP) in a dose-dependent manner, as well as attenuating OS by decreasing markers such as MDA and increasing SOD activity. ASX also improved endothelial function by increasing the bioavailability of NO, which plays a crucial role in vasodilation and blood pressure regulation [88]. In addition, ASX promoted vascular remodeling, reducing arterial stiffness and aortic wall thickness in hypertensive rats. Clinical studies have shown that ASX can improve the lipid profile, reduce visceral body fat and lower blood pressure in patients with T2DM [90].

Carotenoids, such as lycopene, can inhibit angiotensin II, which promotes the constriction of blood vessels and raises blood pressure. This helps reduce blood pressure and improves cardiovascular health [88].

Several marine algae, such as *Ascophyllum nodosum* and *Fucus vesiculosus*, stand out for their potential in the management of MetS. These algae, found in the nutraceutical Gdue, are rich in bioactive compounds such as polyphenols, sulphated polysaccharides, carotenoids and polyunsaturated fatty acids, which have beneficial effects on lipid and carbohydrate metabolism [74]. These compounds act by inhibiting enzymes such as α-amylase and α-glucosidase, which reduces the absorption and digestion of carbohydrates, contributing to their antidiabetic and antihyperglycemic action [91,92].

The polyphenols and sulphated polysaccharides present in these algae have shown significant effects on blood glucose control and insulin sensitivity. Clinical studies suggest that consuming extracts of these algae can improve the insulin response and reduce postprandial glucose and insulin levels [92]. Ascophyllum nodosum extract, for example, has been effective in reducing plasma glucose, while Fucus vesiculosus can improve glycemic response, particularly in women [92]. The study by Martin et al. involved 50 patients aged between 18 and 60. In the study, the participants were given a combination of *Ascophyllum nodosum*, *Fucus vesiculosus* and chromium picolinate for a period of 6 months. The route of exposure was oral, and the participants took three capsules a day, each containing 237.5 mg of *Ascophyllum nodosum*, 12.5 mg of *Fucus vesiculosus* and 7.5 μg of chromium picolinate. The study was a randomized, placebo-controlled clinical trial, and the outcomes assessed included anthropometric parameters, glycemic control and lipid profile. Waist circumference decreased significantly after 6 months of treatment, indicating that the majority of participants experienced a reduction in this parameter. In addition, blood glucose and insulin levels were also significantly reduced after treatment. The Homeostasis Model Assessment (HOMA) index, which assesses insulin sensitivity, showed a significant decrease, suggesting an improvement in the state of insulin sensitivity [92,93].

In addition to their antihyperglycemic properties, *A. nodosum* and *F. vesiculosus* are rich in fucoidans, which have remarkable anti-inflammatory and antioxidant effects. These compounds are capable of modulating inflammatory biomarkers such as CRP and TNF-α, both of which are closely linked to chronic inflammation associated with MetS [92]. The ability to reduce these inflammatory markers is particularly important since systemic inflammation is a crucial factor in the development of CVD and IR, two central components of MetS.

Fucoidans have also been associated with cardiovascular benefits, including the inhibition of angiotensin-converting enzyme, which is responsible for regulating blood pressure. The presence of alginates in these algae also contributes to metabolic health, inhibiting the absorption of lipids and promoting satiety, which helps control body weight [91,92].

The positive impact of these algae on metabolic health is not limited to glycemic and lipid control. *Ascophyllum nodosum* and *Fucus vesiculosus* also influence the composition of the intestinal microbiota, promoting the growth of beneficial bacteria that affect digestion and nutrient absorption. Modulating the microbiota can thus improve insulin sensitivity and reduce intestinal inflammation, another relevant factor in controlling MetS [92].

Moreover, compounds like phlorotannins and fucoidans in brown seaweeds (*Ecklonia cava* and *Sargassum hystrix*) have been shown to improve insulin sensitivity and glycemic control, contributing further to MetS management [94].

*Gelidium elegans* (GEE), an edible red alga, has shown therapeutic potential in reducing obesity markers and regulating metabolic biomarkers. In a study, 78 participants were randomized, with those in the test group receiving three tablets of GEE extract (1000 mg/day) once a day for 12 weeks and those in the placebo group receiving placebo for 12 weeks on the same regimen as the test group. The results showed that the group treated with GEE showed significant reductions in body weight and total fat mass compared to the placebo group. In addition, GEE significantly reduced abdominal visceral fat, an important marker of obesity and the risks associated with MetS. Although the changes in TG levels were not statistically significant, there was a trend towards a reduction in these levels in the treated group. GEE was also shown to be effective in modulating adipocytes, reducing the expression of adipogenic transcription factors such as PPAR-γ and C/EBP-α, which are crucial in the development and storage of fat cells [95]. This suggests that *G. elegans* may inhibit adipocyte differentiation and fat accumulation in humans, similar to the effects observed in animal studies.

*Undaria pinnatifida* (wakame), a brown alga, has attracted attention for its beneficial effects on glycemic control and lipid profile. A 4-week supplementation study in patients with T2DM showed a significant increase in total dietary fiber intake, along with reductions in fasting and postprandial glucose levels, TG and LDL-C [81]. In addition, improvements in antioxidant enzyme activities were observed. In a separate study involving premenopausal women with NAFLD, *U. pinnatifida* supplementation resulted in a decrease in body weight, liver fat, serum TG and inflammation markers, indicating its potential role in the management of metabolic conditions. Other research has also confirmed wakame’s ability to reduce postprandial glucose levels and improve blood pressure in hypertensive patients [81]. A study analyzed the clinical effects of *U. pinnatifida* on blood pressure in hypertensive patients. Patients were assigned either to the treatment group, which received 5 g of dry wakame powder packaged in 12 capsules, which were taken daily in three doses divided with meals for 8 weeks, or the control group, which did not receive the supplement. The main results of the study indicate that the group taking wakame showed a significant reduction in blood pressure. After 4 weeks, SBP fell by 13 mmHg and DBP by 9 mmHg. After 8 weeks, the reductions were 8 mmHg for both SBP and DBP. In the control group, there were no significant changes. In addition, patients with hypercholesterolemia in the wakame group also showed an 8% reduction in TC after 4 weeks of treatment [96].

Similarly, *Saccharina japonica* (formerly *Laminaria japonica*) and other *Laminaria/Saccharina* species are recognized for their health benefits. One study investigated the effects of supplementing fucoxanthin, a carotenoid found in seaweed, especially *Laminaria japonica* (kombu), in Japanese adults with mild obesity. Conducted as a randomized, double-blind, placebo-controlled clinical trial, 33 participants with a BMI ≥ 25 and <30 kg/m^2^ were divided into groups that received doses of 1 mg or 3 mg of fucoxanthin per day administered orally in capsules over a period of 4 weeks. The results showed that the 3 mg/day dose led to a significant reduction in body weight, BMI and visceral fat area compared to the placebo group, while the 1 mg/day dose significantly reduced subcutaneous fat [97].

In addition, fucoidan, a polysaccharide derived from *Saccharina/Laminaria*, has been associated with improvements in insulin secretion and resistance, as well as reductions in DBP and LDL-C levels. *Saccharina*/*Laminaria*’s antioxidant properties were further confirmed in a study where fermented *S. japonica* improved the liver’s antioxidant defenses, leading to reductions in OS markers such as MDA [81].

Another study examined the effects of the edible brown alga *Sargassum horneri* on high-fat (HF) diet-induced obesity, diabetes and hepatic steatosis in mice. The study used six-week-old male C57BL/6J mice divided into four groups: normal diet group, HF diet group, HF group supplemented with 2% *Sargassum horneri* (ShL) and HF group supplemented with 6% *Sargassum horneri* (ShH). The treatment lasted 13 weeks, with the seaweed mixed directly into the mice’s diet, and the route of exposure was oral. The results showed that supplementation with *Sargassum horneri* significantly reduced weight gain and fat accumulation in adipose tissue and the liver, as well as decreasing high blood glucose levels. The seaweed also improved IR and increased the fecal excretion of TG, suggesting that its action is due to the inhibition of the intestinal absorption of fats. This effect was confirmed by in vitro tests, which showed that *Sargassum horneri* inhibits the activity of pancreatic lipase, an enzyme essential for the digestion of fats [98]. The bioactive components, such as alginate, fucoidan and fucoxanthin, seem to be mainly responsible for the beneficial effects observed. However, it is important to note that despite the promising results, further studies in humans are needed to validate these effects.

*Durvillaea antarctica*, a brown alga widely consumed in Chile, has been highlighted for its ability to promote cardiometabolic health and modulate the composition of the intestinal microbiota. Studies have shown that the dietary fiber present in *Durvillaea antarctica* plays an important role in reducing levels of LDL-C and plasma TG, which can help reduce cardiovascular risk. In addition, the presence of sulphated polysaccharides, such as fucoidans and laminarins, has demonstrated significant immunomodulatory and anti-inflammatory effects, being able to modulate the production of pro-inflammatory cytokines such as IL-6 and TNF-α [99]. By improving the lipid profile and reducing systemic inflammation, these compounds contribute directly to the regulation of MetS biomarkers.

A key aspect of *Durvillaea antarctica*’s benefits is its influence on the intestinal microbiota. The polysaccharides present in the algae act as prebiotics, promoting the growth of beneficial bacteria such as Bacteroidetes and Firmicutes, which play a role in modulating lipid and glucose metabolism. This modulation is particularly relevant for individuals with MetS, as it can improve insulin sensitivity and promote glycemic homeostasis by regulating glucose absorption and the insulin response. Another important factor is the presence of the carotenoid fucoxanthin in *Durvillaea antarctica*, which has been shown to inhibit lipogenesis and promote lipolysis, favoring a reduction in body fat and an improvement in the glycemic profile. Studies also indicate that fucoxanthin can modulate the expression of adiponectin and leptin, hormones involved in regulating body weight and lipid metabolism [99,100]. Another study evaluated the polyphenolic extract of the alga *Durvillaea antarctica* (cochayuyo) as an inhibitor of the digestive enzymes α-glucosidase and α-amylase, which are essential for controlling postprandial glycemia in T2DM. Collected in the south of Chile, the seaweed had its extracts obtained by acetone extraction and by hot pressurized liquid extraction with ethanol/water (50%). Tests at concentrations of 1 µg/mL to 2000 µg/mL showed that the acetone extract inhibited almost 100% of α-glucosidase activity at 1000 µg/mL, outperforming acarbose (a commercial inhibitor), which reduced activity by 40.4%. The ethanolic extract was also effective, with activity reduced to 3.1%. The inhibition of α-amylase was moderate, with the acetone extract inhibiting 56.6% at 2000 µg/mL. These results highlight the potential of *Durvillaea antarctica* extract as a promising candidate for the development of functional ingredients that help control postprandial glycemia [101].

Collectively, brown algae, including *Ascophyllum nodosum*, *Fucus vesiculosus*, *Ecklonia cava*, *Sargassum hystrix*, *Undaria pinnatifida*, *Saccharina japonica*, *Sargassum horneri* and *Durvillaea antarctica*, have significant antidiabetic potential due to their bioactive compounds, including phlorotannins, fucoidans, laminarins and alginates. These compounds positively influence glucose metabolism, increase insulin sensitivity and reduce OS, contributing to the management of MetS and related diseases.

The red alga *Gloiopeltis furcata* is traditionally used in Asian cuisine and valued for its medicinal properties. Rich in sulphated polysaccharides such as funoran, this alga has been shown to be effective in combating obesity and diabetes, conditions often associated with weight gain, IR and chronic inflammation. In one study, male mice of the C57BL/6J strain were used as an experimental model to investigate the antiobesity and antidiabetic effects of the edible alga *Gloiopeltis furcata*. Exposure to the algae was through feeding a high-fat diet (60% of energy coming from fat) supplemented with 2% or 6% *G. furcata* for 13 weeks. Supplementation with *G. furcata* resulted in a significant reduction in weight gain, body fat accumulation and glucose and cholesterol levels. In addition, the algae stimulated the excretion of fat in the feces and reduced inflammation and OS, crucial factors in the prevention of metabolic diseases [102].

Similarly, the alga *Campylaephora hypnaeoides* has also shown great potential in the treatment of metabolic disorders such as obesity and diabetes. A study analyzed the effects of the edible red alga *Campylaephora hypnaeoides* on mice fed a high-fat diet, with the aim of investigating its impact on obesity and related metabolic disorders such as diabetes. The algae were administered in concentrations of 2% and 6%, incorporated directly into the diet of the mice, which consumed it orally over 13 weeks. Male mice of the C57BL/6J strain were used as an experimental model, divided into four groups: one group with a normal diet, one group with an HF diet and two groups with an HF diet supplemented with 2% and 6% *C. hypnaeoides*. Supplementation with this alga resulted in a significant reduction in weight gain, IR and blood glucose levels. In addition, *C. hypnaeoides* showed an antioxidant and anti-inflammatory effect, helping to reduce OS and inflammation in animals, as observed with *G. furcata* [103].

Both algae, *Gloiopeltis furcata* and *Campylaephora hypnaeoides*, have a positive impact on lipid metabolism, inhibiting fat absorption and promoting the fecal excretion of lipids. Studies suggest that these algae may be effective as natural dietary interventions in the prevention and treatment of metabolic disorders such as obesity, diabetes and hepatic steatosis, thanks to their antiobesity, antioxidant and anti-inflammatory effects [102,103]. Although clinical investigations are necessary to confirm its effectiveness in humans.

Considering the length of this manuscript and the need to present the information clearly and concisely, the authors decided to organize the data in Table 2 and Figure 3. This approach aims to facilitate visualization and understanding of the information discussed throughout the text.

## 7. Conclusions

This review highlighted several potential biomarkers associated with MetS, specifically exploring the lipid profile, inflammation, OS and miRNAs. A clear association was observed between lipid profile parameters, such as elevated TG levels, reduced LDL-C and increased TG/HDL ratio, and the presence of MetS. The ApoB/ApoA1 ratio is considered a more accurate indicator for assessing the balance between atherogenic and antiatherogenic lipoproteins and is an ideal marker for lipid alterations associated with IR and MetS. These lipid abnormalities are closely linked to an increased risk of CVD and T2DM, highlighting their importance as predictive markers and therapeutic targets.

Systemic inflammation, evidenced by increased levels of pro-inflammatory cytokines such as IL-6 and TNF-α, together with biomarkers of endothelial dysfunction such as PAI-1 and CRP, demonstrate a chronic inflammatory state present in MetS. This inflammation is related to IR, dyslipidemia and obesity, contributing to the development of cardiovascular and metabolic complications. Reduced levels of adiponectin are associated with IR and inflammation, while high levels of leptin reflect obesity and promote inflammation.

Higher levels of leptin/adiponectin are associated with a greater risk of developing MetS and are a more accurate marker than adiponectin or leptin alone. Omentin, which has anti-inflammatory properties, tends to decrease in this condition. There is a positive correlation between fetuin-A and MetS and a negative correlation between adiponectin and MetS, with the fetuin-A/adiponectin ratio being a more sensitive indicator for assessing MetS than these two parameters alone.

OS biomarkers, such as lipid peroxides, protein and nucleic acid oxidation, are increased in individuals with MetS, while antioxidant defenses, such as SOD and antioxidant vitamins, are decreased. The presence of LDL-ox is associated with IR and the pathogenesis of MetS. Enzymatic and non-enzymatic markers, such as GGT and uric acid, are indicative of OS and are correlated with the severity of MetS and the risk of CVD.

In addition, the analysis of miRNAs revealed distinct changes in their expression in individuals with MetS, showing their regulatory role in dysfunctional metabolic processes and associated chronic inflammation. MiRNAs such as miR-15a-5p, miR5-17-5p and miR-24-3p have been identified as important predictors in the presence of MetS, sharing genes involved in the regulation of metabolic pathways, including insulin and fatty acid metabolism. Other examples include overexpressed miRNAs, such as miR-142-3p, miR-140-5p, miR-222, miR-143 and miR-130, which are positively correlated with BMI, IR and lipid profile parameters, while underexpressed miRNAs, such as miR-532-5p, miR-423-5p, miR-520c-3p, miR-146a and miR-15a, are inversely correlated with these same parameters. These examples highlight the diversity of miRNAs involved in regulating the pathophysiology of MetS and suggest their potential as metabolic biomarkers and therapeutic targets. However, further studies are needed in this field to fully understand their contribution and clinical applicability.

The combination of biomarkers has the potential to substantially improve sensitivity and specificity in the early detection and prediction of MetS. An integrated approach using multiple biomarkers can increase diagnostic accuracy, allowing for more refined risk stratification and more targeted treatment. By identifying individuals at greater risk of developing cardiovascular and metabolic complications, this approach facilitates more effective interventions, resulting in a reduction in the morbidity and mortality associated with MetS.

In addition, the inclusion of dietary interventions based on functional foods, such as seaweed, offers a promising strategy for managing MetS. Seaweed, rich in bioactive compounds with antioxidant, anti-inflammatory and antidiabetic properties, can complement traditional therapeutic approaches, improving metabolic parameters and contributing to the prevention and treatment of MetS. For example, *Undaria pinnatifida* has been shown to reduce glucose and lipid levels in patients with T2DM, while *Saccharina japonica* has fucoxanthin, which helps to reduce body weight and abdominal fat. *Ascophyllum nodosum* and *Fucus vesiculosus*, present in nutraceuticals, show benefits in controlling carbohydrate and lipid metabolism. Algae such as *Saccharina japonica* and *Ecklonia cava* offer antioxidant and anti-inflammatory benefits that help reduce OS and regulate glucose. Moreover, species such as *Gloiopeltis furcata* and *Campylaephora hypnaeoides*, both red algae, have shown potential in combating obesity and diabetes by reducing lipid absorption and promoting fat excretion. The growing evidence on the benefits of algae in metabolic health highlights the importance of integrating these foods into the diet, with a view to more effective management of MetS and its comorbidities.

We recognize some limitations. These include gaps in the current understanding of certain biomarkers related to seaweed consumption and challenges in translating the results into effective clinical applications. To overcome these barriers, we suggest that future research focus on areas such as conducting large-scale clinical trials that can validate the efficacy and safety of seaweed-based dietary interventions in diverse populations; investigating the synergistic effects of seaweed bioactives with other therapeutic approaches, such as drugs or dietary supplements; and in-depth studies on the long-term impacts of seaweed-rich diets, with special attention to sustainability and the potential for personalization in MetS management. Future studies are needed to fully understand the efficacy and mechanisms behind these improvements, but incorporating algae into the diet could represent a promising preventive and therapeutic approach.

These advances will not only benefit affected individuals but also have a positive impact on public health, helping to mitigate the growing global burden of MetS.

## Figures and Tables

**Figure 1 marinedrugs-22-00550-f001:**
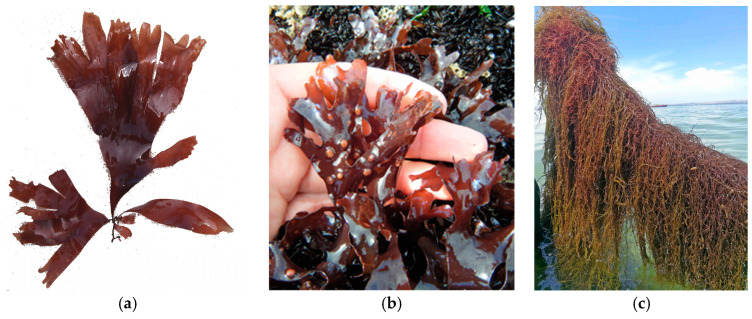
(**a**) *Palmaria palmata*; (**b**) *Chondrus crispus*; (**c**) *Gracilariopsis longissima*.

**Figure 2 marinedrugs-22-00550-f002:**
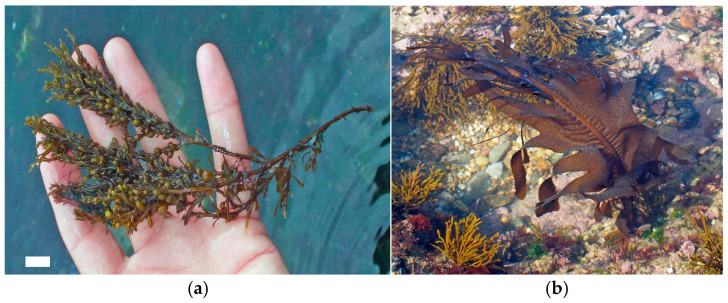
(**a**) *Sargassum muticum*; (**b**) *Undaria pinnatifida*.

**Figure 3 marinedrugs-22-00550-f003:**
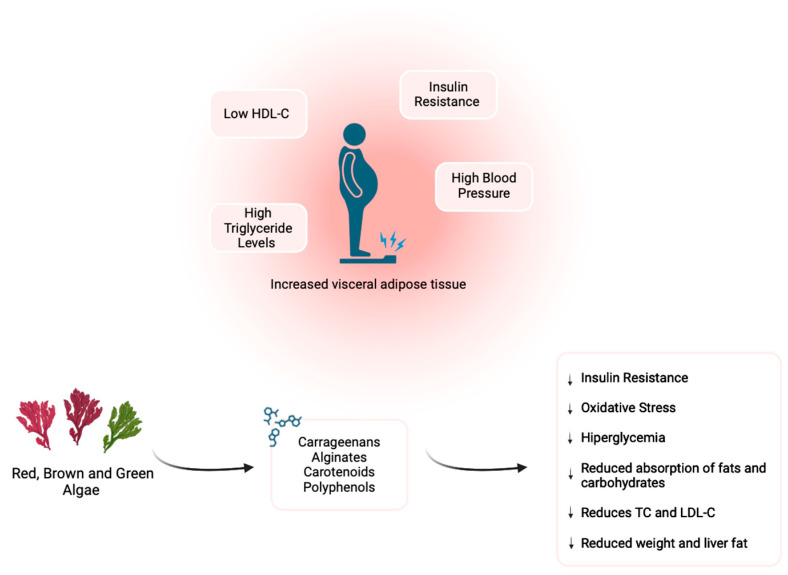
Summary of the benefits of algae in Metabolic Syndrome.

**Table 1 marinedrugs-22-00550-t001:** Summary of the studies used in this article on the relationship between biomarkers and Metabolic Syndrome.

Biomarker	Population	Conclusion of the Study	Reference
TG/HDL-C	1267 individuals aged ≥ 65 years, 234 with MetS and 1033 without MetS 524 children aged between 10 and 16 with obesity	High TG/HDL-C ratio suggests higher risk of MetS.	Nie et al. [31]; Nur Zati Iwani et al. [30]
ApoB/ApoA1	108 individuals, 54 with MetS and 54 without MetS	ApoB/ApoA1 increased significantly with the increase in the number of MetS components.	Kir et al. [32]
IL-6	108 individuals, 54 with MetS and 54 without MetS	High levels of IL-6 were observed in individuals with MetS.	Kir et al. [32]
TNF-α	160 adults between the ages of 55 and 80, 80 with MetS and 80 without MetS 96 individuals aged ≥ 60 years, 59 with MetS and 37 without MetS	Higher levels of TNF-α have been observed in MetS patients compared to individuals without MetS and have also been associated with MetS risk factors such as obesity, IR and T2DM.	Monserrat-Mesquida et al. [33]; Tylutka et al. [34]
PAI-1	379 individuals, aged between 30 and 79, 294 with MetS and 85 without MetS	PAI-1 levels were significantly higher in individuals with MetS compared to individuals without MetS. This increase was associated with the risk of MetS and adverse effects on MetS components.	Nawaz et al. [35]
CRP	107 children and adolescents aged 6 to 18; 21 were normal weight, 22 were overweight/obese without MetS, and 64 were overweight/obese with MetS	Serum CRP levels were higher among overweight/obese children with MetS.	Cura-Esquivel et al. [36]
Leptin/adiponectin	2691 adults, 744 with MetS and 1947 without MetS	A higher leptin/adiponectin ratio was positively associated with the risk of developing MetS.	Lee et al. [37]
Omentin	742 teenagers between the ages of 12 and 16	Omentin levels were associated with MetS and central obesity among all MetS components, being negatively correlated with the degree of obesity/overweight. They suggest that omentin level may serve as a predictor of MetS and obesity.	Sun et al. [38]
Fetuin-A/adiponectin	465 elderly individuals, 181 with MetS and 284 without MetS	The fetuin A/adiponectin ratio was significantly associated with all MetS components, showing that this ratio may be a more sensitive index for assessing MetS than fetuin A and adiponectin alone.	Zhou et al. [39]
Ox-LDL	3987 individuals, 393 with MetS and 3594 without MetS	They found that ox-LDL concentrations were associated with MetS and its components. They suggested that it may be a clinically relevant predictor of the development of MetS.	Hurtado-Roca et al. [40]
Uric acid	5758 individuals, 1599 with MetS and 4159 without MetS	High serum uric acid levels have been associated with the risk of MetS.	Jeong et al. [41]
MiR-24-3p	147 children, aged between 6 and 16, 45 with obesity, 52 with obesity and MetS and 50 healthy with normal weight	Expression of miR-24-3p was higher in obese children with MetS. Potential non-invasive marker for children with obesity and with predictive value for the occurrence of MetS.	Zhang et al. [42]
MiR-15a-5p and MiR-17-5p	79 individuals, 39 with MetS and 40 without MetS	MiR-15a-5p and miR-17-5p have been identified as predictive biomarkers of MetS.	Ramzan et al. [43]

MetS: Metabolic Syndrome; TG/HDL-C: triglycerides/high-density lipoprotein cholesterol; ApoB/ApoA1: apolipoprotein B/apolipoprotein A1; IL-6: interleukin 6; TNF-α: tumor necrosis factor alpha; PAI-1: plasminogen activator inhibitor type 1; CRP: C-reactive protein; ox-LDL: oxidized LDL.

**Table 2 marinedrugs-22-00550-t002:** Effects of seaweed on Metabolic Syndrome.

Seaweed/ Compounds	Main Effects	Impact on MetS	Study Effects	References
Carrageenans (red seaweed)	Reduces TC and LDL-C	Improves lipid profile; limited protective effect for MetS due to increase in TG	Study with 30 hypercholesterolemic subjects (20–64 years), 100 mL/day of vegetable jelly with carrageenan for 60 days.	[82,83]
Alginates (brown seaweed)	Reduced absorption of fats and carbohydrates, blood glucose control, feeling of satiety	Helps reduce postprandial glucose, cholesterol and body weight; significant effects on CVD prevention and MetS control	Randomized controlled clinical study conducted over 8 weeks with 50 Japanese participants with a BMI ≥ 25 and <30 kg/m^2^, administered a daily supplement of 3 g of *Saccharina japonica* alginate (iodine-reduced algae powder).	[84,87]
Carotenoids (astaxanthin)	Potent antioxidants, they reduce glucose and improve blood pressure, endothelial function and vascular remodeling	They help control blood pressure, improve blood glucose and lipid profile and prevent cardiovascular complications	8-week randomized, placebo-controlled clinical trial in patients with T2DM, with a dose of 8 mg/day of ASX.	[88,89]
*Ascophyllum nodosum* e *Fucus vesiculosus*	Blood glucose control, improved insulin sensitivity, anti-inflammatory and antioxidant effects	Reduces glucose, improves insulin sensitivity and lipid profile, helps prevent chronic inflammation and CVD	Randomized controlled clinical study for 6 months with 50 participants (18–60 years), with 3 capsules/day (237.5 mg of *A. nodosum*, 12.5 mg of *F. vesiculosus* and 7.5 μg of chromium picolinate).	[92,93,104]
*Gelidium elegans*	Reducing body weight and visceral fat and regulating adipogenesis	Anti-obesity potential and visceral fat control, positive effect on MetS control	78 randomized participants, testing 1000 mg/day of *Gelidium elegans* for 12 weeks, compared to placebo.	[95]
*Undaria pinnatifida* (wakame)	Improved blood glucose and lipid profile, reduced weight and liver fat, increased antioxidants	Benefits in controlling glucose, cholesterol and obesity; antioxidant effects prevent cardiovascular damage	Clinical study with hypertensive patients, 5 g wakame powder/day for 8 weeks, comparison with control group.	[81,96]
*Saccharina japonica* (fucoxanthin)	Reduced body weight, improved IR, lowered LDL-C and blood pressure	General metabolic benefits; contributes to the management of blood glucose, lipids and blood pressure	Randomized, double-blind, placebo-controlled clinical study with 33 Japanese adults (BMI 25–30 kg/m^2^), doses of 1 mg and 3 mg/day for 4 weeks.	[81,97]
*Sargassum horneri*	Weight reduction, improvement in IR, fecal excretion of TG, inhibitory action on lipase	Prevention of obesity and diabetes, improvement of lipid profile and IR; general effects against MetS	Study with male C57BL/6J mice, HF diet with 2% or 6% *Sargassum horneri* for 13 weeks.	[98]
*Durvillaea antarctica*	Reduction of LDL-C and TG, modulation of intestinal microbiota, anti-inflammatory and antioxidant action	Positive effect on lipid and inflammatory control, helps control weight and blood glucose, contributes to the prevention of MetS	Study evaluated the polyphenolic extract of *Durvillaea antarctica*. The acetone extract inhibited 100% of α-glucosidase at 1000 µg/mL and reduced α-amylase activity by 56.6% at 2000 µg/mL.	[99,101]
*Gloiopeltis furcata*	Weight reduction, improvement of blood glucose and cholesterol, anti-inflammatory and antioxidant action	Helps prevent obesity and diabetes; antioxidant and anti-inflammatory effect relevant to MetS	Study with male C57BL/6J mice, high-fat diet (60% of energy from fat) with 2% or 6% *G. furcata* for 13 weeks.	[102]
*Campylaephora hypnaeoides*	Reduction of visceral fat, improvement of IR and antioxidant action	Helps control obesity and inflammation; positive impact on the treatment of metabolic disorders	Study with C57BL/6J male mice, HF diet with 2% and 6% *C. hypnaeoides* for 13 weeks, compared with normal diet and HF.	[103]

MetS: Metabolic Syndrome; TC: Total Cholesterol; LDL-C: Low-Density Lipoprotein Cholesterol; TG: Triglycerides; CVD: Cardiovascular Disease; ASX: astaxanthin; IR: Insulin Resistance; HF: high-fat.

## Data Availability

Not applicable.

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
