# Peer review of "Biomarkers and Seaweed-Based Nutritional Interventions in Metabolic Syndrome: A Comprehensive Review"

_marinedrugs, 2024, doi:10.3390/md22120550_

Round 1
Reviewer 1 Report
Comments and Suggestions for Authors
This paper seems to be well organized by subtopic. However, there are a few grammatical errors.
In the abstract, "Early identification and effective monitoring of MetS are crucial" could be improved by changing to "is crucial" to match the singular concept implied by "identification and monitoring".
In the introduction part, "he development of pathologies such as type 2 diabetes mellitus," "he" should be corrected to "the".
Also, terms like "Non-alcoholic fatty liver disease" should consistently use "NAFLD" after the first mention. Consistency in acronym usage enhances readability, especially in scientific texts.
Reviewer 2 Report
Comments and Suggestions for Authors
MS is a term given to a set of conditions, including hyperglycemia, atherogenic dyslipidemia, insulin resistance (IR), systemic arterial hypertension and central/abdominal obesity. Te current review summarized the relevant biomarkers of MetS and explores how algae-based nutritional interventions can positively impact these parameters, offering new perspectives for the management of this complex condition. Overall,the topic is interesting and the paper is well written. I have some concerns.
1) Several novel biomarkers for MS were selected and introduced, like inflammatory markers, and microRNAs. How these biomarkers were decided by the authors? Cause many other factors are associated with MS, such as LPS, IL-1β, uric acid, and gut microbiota disorders. This issue should be clearly introduced.
2) The category of Macroalgae and their influence were poorly organized. Please properly re-organize sections “5. Macroalgae” and “6. The seaweed-based dietary interventions for metabolic syndrome” by using tables, figures and sub-titles.
3) It’s better to combine the images of seaweed into one large figure.
4) “POSSIBLE BIOMARKERS OF METABOLIC SYNDROME”
Reviewer 3 Report
Comments and Suggestions for Authors
Author Response
Please see the attachment.

Round 2
Reviewer 3 Report
Comments and Suggestions for Authors
Authors have answered all questions and the manuscript could be accepted.